# ECG performance in simultaneous recordings of five wearable devices using a new morphological noise-to-signal index and Smith-Waterman-based RR interval comparisons

**Dominic Bläsing**[1,2☯]*, **Anja Buder**[3], **Julian Elias Reiser**[4], **Maria Nisser**[3], **Steffen Derlien**[3], **Marcus Vollmer**[5,6☯]

**1** Institute of Psychology, University Greifswald, Greifswald, Germany, **2** Institute for Community Medicine, Prevention Research and Social Medicine, University Medicine Greifswald, Greifswald, Germany, **3** Institute of Physiotherapy, University Hospital Jena, Jena, Germany, **4** Leibniz Research Centre for Working Environment and Human Factors – IfADo, Dortmund, Germany, **5** Institute of Bioinformatics, University Medicine Greifswald, Greifswald, Germany, **6** German Centre for Cardiovascular Research (DZHK), partner site Greifswald, Greifswald, Germany

☯ These authors contributed equally to this work.
* dominic.Blaesing@med.uni-greifswald.de

**Data Availability Statement:** Raw data from all devices and processed data generated during the analysis of the current study have been published

## Abstract

### Background

Numerous wearables are used in a research context to record cardiac activity although their validity and usability has not been fully investigated. The objectives of this study is the cross-model comparison of data quality at different realistic use cases (cognitive and physical tasks). The recording quality is expressed by the ability to accurately detect the QRS complex, the amount of noise in the data, and the quality of RR intervals.

### Methods

Five ECG devices (eMotion Faros 360˚, Hexoskin Hx1, NeXus-10 MKII, Polar RS800 Multi and SOMNOtouch NIBP) were attached and simultaneously tested in 13 participants. Used test conditions included: measurements during rest, treadmill walking/running, and a cognitive 2-back task. Signal quality was assessed by a new local morphological quality parameter morphSQ which is defined as a weighted peak noise-to-signal ratio on percentage scale. The QRS detection performance was evaluated with eplimited on synchronized data by comparison to ground truth annotations. A modification of the Smith-Waterman algorithm has been used to assess the RR interval quality and to classify incorrect beat annotations. Evaluation metrics includes the positive predictive value, false negative rates, and F1 scores for beat detection performance.

### Results

All used devices achieved sufficient signal quality in non-movement conditions. Over all experimental phases, insufficient quality expressed by morphSQ values below 10% was

**Funding:** DB acknowledges the financial support by the Federal Ministry of Education and Research of Germany in the project Montexas4.0 (FKZ 02L15A261 https://www.bmbf.de/bmbf/en/home/home_node.html), MV acknowledges travel grants from the German Centre for Cardiovascular Research (DZHK, partner site Greifswald https://dzhk.de/en/). The funders had no role in study design, data collection and analysis, decision to publish, or preparation of the manuscript.

**Competing interests:** The authors have declared that no competing interests exist.

only found in 1.22% of the recorded beats using eMotion Faros 360˚whereas the rate was 8.67% with Hexoskin Hx1. Nevertheless, QRS detection performed well across all used devices with positive predictive values between 0.985 and 1.000. False negative rates are ranging between 0.003 and 0.017. eMotion Faros 360˚achieved the most stable results among the tested devices with only 5 false positive and 19 misplaced beats across all recordings identified by the Smith-Waterman approach.

## Conclusion

Data quality was assessed by two new approaches: analyzing the noise-to-signal ratio using morphSQ, and RR interval quality using Smith-Waterman. Both methods deliver comparable results. However the Smith-Waterman approach allows the direct comparison of RR intervals without the need for signal synchronization whereas morphSQ can be computed locally.

## Introduction

Miniaturization and commercialism of physiological measurement in combination with the ongoing trend for self-quantification of personal health and fitness lead to a new variety of professional or semi-professional measurement devices [1–4]. Fitness trackers, smart watches, and even clothing with bio-physiological sensors were developed to help people gather data for health related aspects such as physical activity, sleep phases, heart rate (HR), cardiac arrhythmias, or to measure stress [2, 5]. A rising number of those devices are also used in the research area [6], with an appropriate methodology to verify their validity and suitability in those contexts still missing [7].

### ECG fundamentals

HR and heart rate variability (HRV) analyses deduced from ECG signals have become important in many scientific research areas (psychology, occupational science, telemedicine, sports science) as well as in the public domain. To obtain reliable and valid results for HR and HRV analysis, high signal quality and heart beat detection accuracy are indispensable.

In medicine, the 12-lead ECG is the diagnostic gold standard for a comprehensive and meaningful diagnosis in a stationary setting [8]. As an alternative approach, Norman J. Holter invented a method to measure ECGs in an ambulatory setting for time spans up to 24 hours [9]. Compared to a 12-lead ECG, the Holter ECG normally corresponds closest to the V5 or V1 leads [10]. The biggest advantage is the ability to measure an ECG in field trials and during everyday life. For a non-clinical use case in field or laboratory, the ambulant setting allows the participants to move freely and interact naturally with their surroundings.

At the moment most commercially available wearable devices rely on the usage of photo-plethysmography (PPG) rather than writing a real ECG. Modern mobile ECG measurement solutions are often limited due to their reduced sampling frequency (below 1000 Hz) or their decreased ability for continuous measurement and free movement (restricted to resting measurement or momentary assessment). Future developments, especially in the area of wireless ECG technology, seem to be promising [11, 12].

Using an ECG, it is possible to record the accumulated changes over time in electrical cell potentials over different heart muscle cells that results in the typical PQRST morphology of a

heartbeat. A normal cardiac cycle can be seen as a sequence of polarization and depolarization of different involved cells starting before the P wave and ending after the T/U wave with the R peak as the highest spot and most prominent ECG feature. For clinicians, the whole PQRST sequence is important to detect insufficiency of heart muscle activity itself [10]. In sports, occupational science, psychology, or ergonomics, the intervals between two successive R peaks (RR intervals) are needed to analyze or quantify HR and HRV. Using those interbeat-intervals for further calculations, it is possible to detect changes in physical and/or cognitive workload [13, 14], to quantify training load or general fitness [15, 16], or to obtain clinically relevant information about the functionality of the autonomic nervous system [17]. For most athletes and private users, the ECG signal itself is not of interest. The derived HR or a pre-calculated stress index (some sort of HRV measurement) is what they are interested in.

Beyond clinical applications, ECG measurement techniques using 1 or 2-channel Holter systems, chest straps, or similar non-invasive methods became popular. A variety of devices is used in mobile medicine ranging from watches and in-ear-systems to smart shirts [2, 5]. For future large-scale clinical studies, the vision is to replace standard ECG techniques by mobile technologies benefiting from the ease of use, faster application, and handling [5].

## Aim of this study

High signal quality is necessary in scientific research contexts to guarantee the correctness of the drawn conclusions and derived measures. With an increasing scope of application, the number (and age) of used ECG devices increases, and so does the range of experience in analyzing ECG data. To support the data analysis, many processes can be automated, e.g, heart beat classification, and RR interval determination. The accuracy of the resulting data highly depends on the ECG signal quality.

The aim of this study is to compare usability and data quality of different consumer and professional ECG devices in research and leisure scenarios. Therefore two new approaches to compare data quality are developed. These include a method to assess the amount of noise at a very local level to quantify local disturbances in the waveform of the ECG. The other method is developed for the verification and classification of RR intervals.

## Related work

There are several signal quality indices (SQI) available to compare multiple ECG signals [18, 19], which can be classified as either detectability- (iSQI, bSQI) or artefact-correction-based (pSQI, sSQI, kSQI, fSQI, baseSQI). Detectability-based SQIs usually focus on several ECG leads (or sensors—iSQI) or beat detection algorithms (bSQI) to calculate single indicators. In contrast, artefact-correction-based indices are lead or sensor specific and take several artefact sources into account like muscle artifacts (sSQI), baseline wander (baseSQI), or high frequency sinusoidal noise (kSQI). While many of the indices above require sufficient data for computation, specific methods deliver the local quality of short ECG segments: Such methods analyze the correlation between successive beat cycles (also known as template matching) [20], use spectral information [21], or calculate continuous signal-to-noise ratios (SNR) that are based on local time windows and noise-free signals, e.g., computed through low-pass filtering or wavelet decomposition [22]. In contrast, Hoog Antink et al. [23] defined a shape-based SNR (SNRs) using cardiac templates for the approximation of noise-free signals. Additional methods are summarized in a review article by Satija et al. [24].

Many applied researchers also use concrete HRV indicators or RR intervals to compare the quality among different devices and focus on the assessment of inter-class correlations or Bland-Altman limits of agreement [25–28].

In contrast to the previous studies, a comparison of HRV indicators among the devices is waived due to the fact that those indicators are the results of mathematical allocations of RR intervals. Thus, the crucial point is the validity of RR interval measurement as the consequence of accurate R peak detection. We therefore focus on the overall noise-to-signal ratio (NSR) and on the verification of annotated beats.

## Materials and methods

### Apparatus

For the purpose of investigating the functionality, accuracy, and usability of several ECG-measurement devices, participants were equipped with five sensor systems at the same time. The sensor placement is illustrated in Fig 1. Three clinically certified devices were used, NeXus-10 MKII (Mind Media B.V., NL—NeXus), eMotion Faros 360˚(Mega Electronics Ltd., FI—Faros), and SOMNOtouch NIBP (SOMNOmedics GmbH, GER—SOMNOtouch). In parallel,

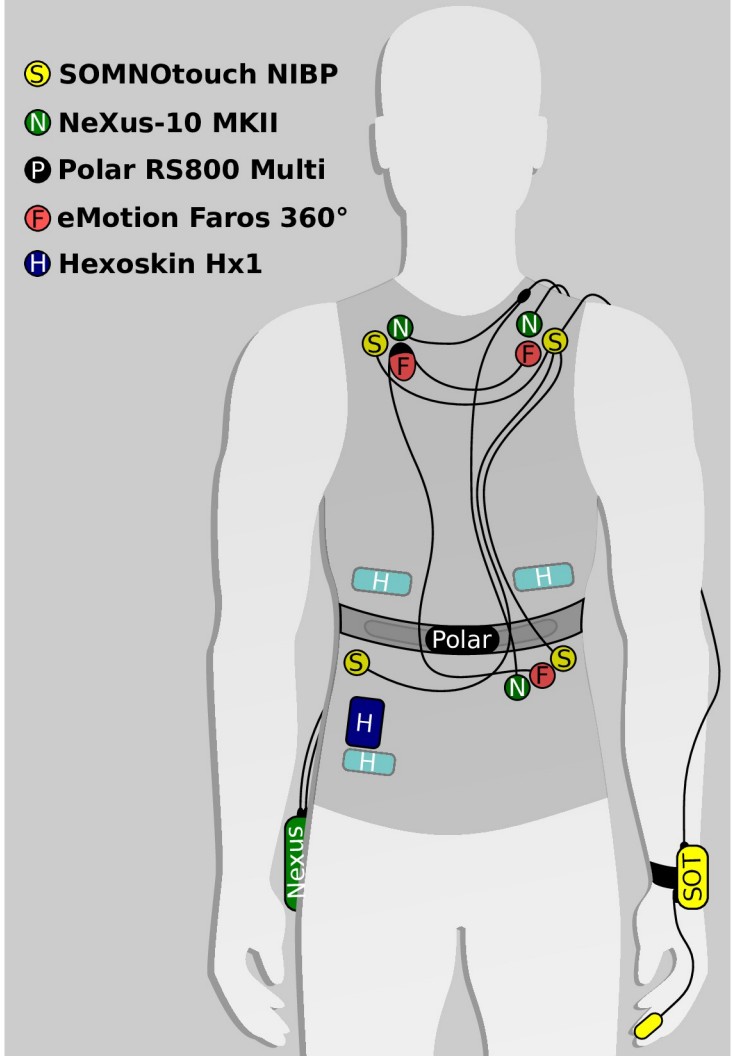

**Fig 1. ECG system configuration for simultaneous measurement (adapted from [34]).**

two consumer products (with a wide and increasing usage in scientific research) were attached: Hexoskin Hx1 (Carré Technologies Inc., CA—Hexoskin), and PolarWatch RS800 Multi (Polar Electro Oy, FI—Polar). All included devices were previously used in field and laboratory settings of the facilities involved. The oldest used deives was the Polar watch, which is, by now, outdated. Newer versions might be able to perform even better in the lower HR range, use better algorithms, or combinations of PPG and chest belts to calculate RR intervals even more precisely and close the gap to the other devices. Despite the fact of a missing raw ECG export functionality, the implementation of wearables in scientific research areas is rising [2]. Adding Polar to the comparison enables us to additionally analyze the quality of RR interval exports, a widely used feature even for consumer products.

Even with medical certifications available for three other included devices there is not always comprehensible evidence for the signal quality. Validation studies are available for Hexoskin [29–31] and Polar [32] with correlation coefficients between 0.8 to 0.9 for Polar and 0.8 to 0.99 for Hexoskin. For SOMNOtouch a validation of the blood pressure function is available but not for the ECG component [33].

To generate physical workload, the participants had to walk on a treadmill (Woodway PPS Med, Woodway GmbH, GER). During the whole experiment every participant had to wear a safety vest connected to an overhead fixation in order to guarantee fall protection.

The electrode placement (Ag/AgCl hydrogel foam electrodes) for ECG measurement was based on the manufacturers' suggestions. Hexoskin is working with textile electrodes integrated in the fabric of a shirt on chest and abdomen. The chest strap of Polar is used for data acquisition, the watch itself is used to store, pre-process, and display data. Table 1 summarizes

**Table 1. General information about the measurement devices.** See S1 Fig for the table with device images.

| | eMotion Faros 360˚ | Hexoskin Hx1 | NeXus-10 MKII | Polar RS800 Multi | SOMNOtouch NIBP |
|---|---|---|---|---|---|
| Manufacturers' page | bittium.com | hexoskin.com | mindmedia.com | polar.com | somnomedics. com |
| Year of production | 2017 | 2015 | 2016 | 2007 | 2015 |
| Follow-up model | Bittium Faros | Hexoskin Smart Shirt | – | Polar Watch V800 + Polar H10 | – |
| Device weight | 15 g | 42 g | 591 g | 47 g | 85 g |
| Additional weight | 16 g (cables) | 145 g (M sized shirt) | 80 g (cables) | 23 g (wearlink) | 32 g + 21 g (cables+$S_pO_2$) |
| Dimensions | 48 x 29 x 12 mm | 40 x 70 x 13 mm | 120 x 140 x 45 mm | watch: 47 x 60 x 15 mm wearlink: 62 x 37 x 12 mm | 74 x 54 x 16 mm |
| Acquisition costs[1] | €€€ (device) | € (device) € (shirt) | €€€€€ (device + accessories) € (ExG sensor) | € (follow-up model) € (wearlink) | €€€€€ (device) € (ExG sensor) € ($S_pO_2$ fingerclip) |
| Medical supplies | ECG electrodes | Skin preparation gel | ECG electrodes | – | ECG electrodes |
| Battery runtime | 24 h (3–channel ECG at 1000 Hz) up to 30 d (RR intervals only) | >14 h (recording mode) up to 400 h (sleep mode) | >24 h (exchangable battery) | >24 h (exchangable battery) | up to 24 h |
| Maximal sampling rate | 1000 Hz | 256 Hz | 8000 Hz | 1000 Hz | 512 Hz |
| ADC precision[2] | 24 bit | 12 bit | 24 bit | NA | 12 bit |

[1] €: ≤500 €; €€: 500–1000 €; €€€: 1000–1500 €; €€€€: 1500–2000 €; €€€€€: >2000 €.

[2] ADC—Analog-to-Digital Converter.

different characteristics of the investigated devices. Product weight was mainly dominated by the storage unit. Not only dimension and weight can be a crucial part of usability for different studies, also the placement of the storage unit (see Fig 1). When it comes to expenses, not only acquisition, but also follow up costs should be considered (see Table 1 for acquisition costs and medical supplies). For NeXus, Faros, and SOMNOtouch additional ECG electrodes are needed. Hexoskin's textile electrodes need conductive gel or glycerin-based cream. While Faros', Hexoskin's and Polar's main focus is ECG measurement, NeXus und SOMNOtouch are built as an integrated solution able to measure different physiological parameters. With regard to Hexoskin, it should also be emphasized that due to the integration of the sensors in a shirt, different sizes are required for the measurements of several subjects. The sampling rate of all devices are in accordance with the necessary parameters suggested by the European Task Force [35] ranging from 256 Hz (Hexoskin) up to 8000 Hz (NeXus). Taking into account the suggestions from Sammito et al. [36], a sampling rate of at least 1000 Hz is considered ideal for the calculation of inter-beat-intervals [36]. Faros, Polar, and NeXus fulfill this requirement. In clinical settings a high sampling rate is prerequisite to detect narrow pacemaker pulses from the resulting ECG [37, 38].

Besides sampling rate, additional technical information is necessary to decide whether the device can be applied in a specific use case (see Table 2). In field tests, for example, weight, battery life and freedom of movement are more important than under laboratory conditions. Certified devices are to be used in clinical studies in which Hexoskin and Polar are not certificated as a medical device on the European market (CE marking). For bio-feedback, NeXus, SOMNOtouch, and Hexoskin provided ECG data in real time, while Polar only showed HR. Faros offers a real-time API to export the live data.

Especially for clinical and research settings, accessibility to the raw data is an important feature most devices offer. For Polar it was only possible to extract pre-calculated RR intervals and no raw ECG. Most devices offered an EDF or EDF+ (European Data Format) export to allow the ease-of-use import into standard analyzing software.

Most devices offer more than a single indicator solution (just ECG/RR) and have, for example, the ability to track additional acceleration data using inertial sensors installed in the storage devices. Optional ExG sensors can be bought for NeXus and SOMNOtouch. Hexoskin was able to measure abdominal and chest breathing and Faros collected additional temperature

**Table 2. Special technical information for scientific studies.**

| | eMotion Faros 360° | Hexoskin Hx1 | NeXus-10 MKII | Polar RS800 Multi | SOMNOtouch NIBP |
|---|---|---|---|---|---|
| Certified medical use | yes | no | yes | no | yes |
| Data transfer | Bluetooth, USB | Bluetooth, USB | Bluetooth, USB | Infrared | Bluetooth, USB |
| Live feedback | yes | yes | yes | yes | yes |
| Export formats[1] | EDF | EDF, WAV, CSV | EDF, EDF+ | txt, hrm | EDF+, RIFF-/ASCII, SCP |
| Raw data[1] | ECG, acceleration, events, temperature | ECG, acceleration, respiration | ECG, acceleration, events | RR intervals | ECG, events |

[1] Used abbreviations: EDF/EDF+—European Data Format; WAV—Waveform Audio File Format; CSV—Comma-Separated Values; hrm—Polar summary export (structured text file); RIFF – Resource Interchange File Format; ASCII – American Standard Code for Information Interchange; SCP—Standard Communication Protocol for Computer assisted electrocardiography (SCP-ECG EN1064:2007).

[2] Note: NeXus and SOMNOtouch can output additional parameters as raw data with additional sensors.

data. Using the finger clip of SOMNOtouch, it is possible to collect data for pulse oximetry and blood pressure.

The subjects has not been equipped with a 12-lead ECG system as a gold standard since the chest leads would have interfered with the textile electrodes of Hexoskin. Clinically validated measuring devices were used in compensation. In addition: The focus of this comparison was not on the application for the detection of clinical abnormalities, but on the application of the devices in ergonomics and sports science, as well as in the leisure context with the focus on the valid and automated detection of R peaks in order to reduce the effort involved in manual follow-up checks.

## Subjects

13 healthy participants (six male) in the age range between 21 and 35 years ($\mu = 28.00$, $\sigma = 4.28$) took part in the experiment. Due to limited sizes of the biometric Hexoskin shirt, only participants could be included for whom a fitting size was available. Further, participants had to confirm that no prior cardiac problems or diseases are known. For both male and female groups, the BMI was in a healthy average range ($\mu_{female} = 22.74$, $\sigma_{female} = 2.40$; $\mu_{male} = 23.66$, $\sigma_{male} = 1.58$). Additionally, 77% had prior experience with treadmill walking. All participants took part in the study without monetary compensation on a voluntary basis. The study was conducted according to the guidelines of the Declaration of Helsinki, and approved by the ethics committee of University of Greifswald (Identifier: BB 171/17, 30$th$ November 2017). Written informed consent to conduct this research has been obtained from the participants.

## Procedure

The experimental setup was divided into four consecutive parts of five minutes each. Task order was identical for each subject to be able to determine time-on-task effects. At first, a baseline resting period was recorded while standing upright on the treadmill (P1). Upon completing P1, participants walked on the treadmillat a moderate speed (1.2 m/s) without treadmill inclination to induce medium, everyday physical workload (P2). The third part consisted of executing a cognitive 2-back task while standing still on the treadmill (P3). During this task a predefined sequence of (randomly generated) numbers between 1 and 6 was presented using a Raspberry Pi 2B (Raspberry Pi Foundation, UK) stored in the backpack and in-ear headphones (QC 2, Bose, USA). All stimuli were of 500 ms length with a 2000 ms inter-stimulus-interval [39]. Participants continuously had to memorize the last two digits and update the information while listening to incoming numbers. After the presentation of each stimulus, participants had to say out loud the number that was heard two stimuli before. Prior to task execution, participants had to pass a one-minute exercise block with at least 70% correctly recalled answers to guarantee right task-execution. As it was the goal to induce a mental state contrary to the correct execution of the task, the correctness of answers was not of interest to the research hypotheses and therefore not quantified. For the final phase (P4), subjects walked uphill on the treadmill with a track inclination of 15% and a speed of 1.2 m/s to induce high physical workload resulting in increased HR, respiratory rate, perceived exertion, and stride length [40, 41]. After each task, participants had to rate their subjective workload on all dimensions of the NASA-TLX [42]. Using the Faros trigger point functionality, timestamps were generated at the beginning of each phase.

## Data preprocessing—Alignment and frequency correction

Necessary preprocessing steps for data alignment were performed as described in previous work [34]. This includes the correction for time shifts as a result of asynchronous record starts

and requires the correction of inaccurate and unsteady sampling frequencies. Alignment was based on the synchronization of heartbeats from a resting period by finding similar patterns in all devices. Sampling frequency was corrected by non-linear resampling to a target sampling frequency of 256 Hz [34].

To improve the automated signal alignment and to further evaluate the detection accuracy of each device, we generated a manual reference annotation of R peaks (beat_ref). All aligned ECG-signals were manually screened and corrected for missing, misplaced, and wrongly identified R peaks arising from the automated heartbeat detection. Following a four-eye-principle, all corrected records were screened a second time. Corrections were performed by experienced research staff. Furthermore, we checked for cardiac arrhythmia to allow the exclusion of affected periods from SQI calculation since arrhythmia would bias the SQI rating due to physiologically based morphological changes. Moreover, arrhythmic periods should be also excluded from HRV computations. Based on the procedure postulated in Vollmer 2017 [43], we identified suspicious segments which were then manually screened. One ectopic beat was identified in the 13 participants leading to our decision to not exclude participants or ECG segments.

Additionally to the use of synchronized data and its complex process to achieve resynchronization, we have extracted RR intervals from raw unsynchronized ECGs by computing the difference of beat locations annotated by eplimited [44]. R peak locations were refined using spline interpolation to increase the resolution of the interval data. This data was then used in comparisons to ground truth intervals computed from the successive differences of beat_ref.

Raw data from all devices and processed data generated during the analysis of the current study have been published open access on PhysioNet [45, 46] and are freely available on PhysioNet (see https://doi.org/10.13026/zhns-t386). Timestamps for experimental stages were manually inspected and relocated by use of movement sensors and HR increase (for phase 2 and 4).

## Statistical analysis

For signal quality analysis, three different approaches were chosen. The first approach focused on the NSR as a general index to quantify the quality of the signal itself (A—Morphological signal quality). The second approach is based on the ability to detect R peaks by comparing identified beat locations with the ground truth annotation (B—Heartbeat detectability). The third approach identifies errors in the automated beat detection process by comparing RR intervals from unsynchronized data (C—Modification of the Smith-Waterman algorithm).

**A—Morphological signal quality.** ECGs measured during different physical activities usually contain distortions such as muscular artifacts, signal baseline wander, and power line inference. To quantify these disturbances around each single R peak, a new local similarity-based morphological signal quality value (morphSQ) is defined. The method belongs the template-based approaches, such as SNRs [23], but in contrast to the available methods it is based on the median cardiac cycle, neglects QRS distortions, and uses a weighting function to increase locality. A short ECG sequence spanning an uneven number of adjacent R peaks is used to calculate the morphSQ value for the current central R peak as illustrated in Fig 2 and described below:

1. There are $k$ [k = 8] full cardiac cycles in the short ECG sequence spanning $k + 1$ R peaks [k + 1 = 9]

2. Calculate the midpoints between consecutive R peaks ($R_i$, $i = 0, \ldots, k$) to define the start ($s_i$, $i = 1, \ldots, k$) and end ($e_i$, $i = 0, \ldots, k - 1$) of k cardiac cycles. The start of the new cardiac cycle $i + 1$ is equal to the end of the previous cycle $i$ ($s_i + 1 = e_i$).

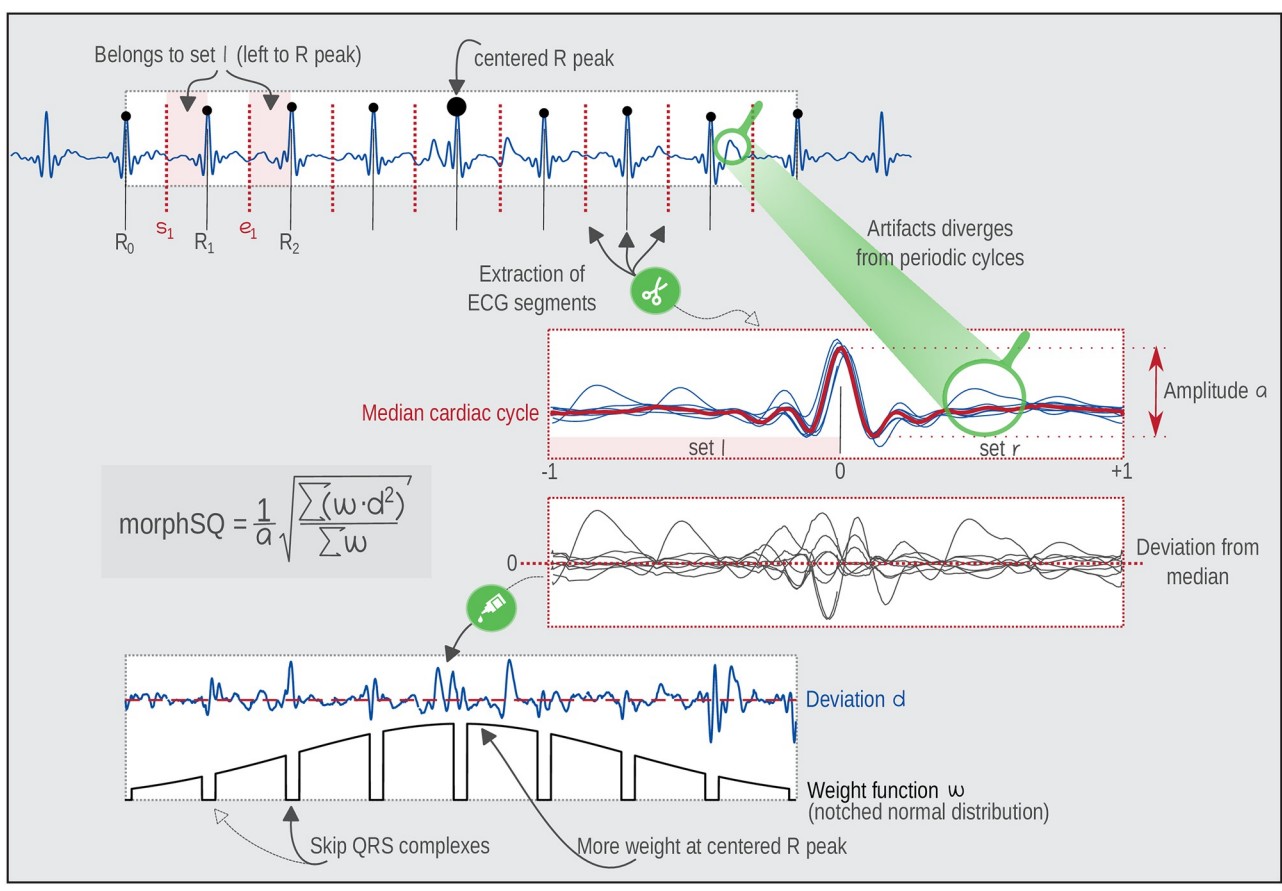

**Fig 2. Illustration of the morphological signal quality process in a sliding window of an ECG (number of cardiac cycles $k = 8$).**

3. Cut each of the cardiac cycles into two sets of segments: The first set (set $l$) contains cycle segments starting at $s_i$ till $R_i(i = 1, \ldots, k)$, which includes the P waves. The second set (set $r$) contains cycle segments starting at $R_i$ till $e_i(i = 0, \ldots, k - 1)$, which includes the T waves.

4. Scale all segments of set $l$ to span the range from -1 to 0.

5. Scale all segments of set $r$ to span the range from 0 to 1.

6. Interpolate each segment linearly to generate values at query points from $-1$ to 1 with 0.002 increment ($-1.00, -0.998, -0.996, \ldots, 1.00$). This allows the direct comparison of the different cardiac cycles at each query point. Optional: Plot all re-scaled segments from $-1$ to 1 to visualize the periodicity of the ECG cycles. QRS templates are pooled around 0.

7. Compute the median cardiac cycle from the overlaid cycles as a robust representative for the observed cardiac cycles and compute the amplitude $a$ of the fit (maximum-minimum).

8. Compute differences $d$ between the observations and the median cycle.

9. Each single difference $d$ will be assigned with a weight $\omega$. More weight is put on cardiac cycles near the central R peak and less weight at the edges of the ECG sequence. We have used weights derived from a normal probability density function from $-2$ to 2.

10. Set weights at the QRS complex ($l_{qrs} = 100$ ms around the R peak) to 0, to not quantify those differences caused by respiration-related modulations of the R peak amplitude [47].

11. morphSQ is defined as the weighted sum of squared differences normalized by the local amplitude of the ECG (Eq 1):

$$\text{morphSQ} = \frac{1}{a} \sqrt{\frac{\sum \omega d^2}{\sum \omega}} \tag{1}$$

The used specifications for the parameters $k$, $\omega$, $l_{qrs}$, as well as the used cut-off-value of 10% for morphSQ were chosen based on theoretical assumptions and mathematical necessities. Setting $k = 8$ and weights $\omega$ taken from notched Gaussian distribution function were necessary to reinforce the locality of morphSQ. Increasing $k$ or making $\omega$ constant would lead to an reduced influence of the momentarily centered beat (R peak). Considering the physiology basis of the QRS complex a notching length of 100 ms was chosen for $l_{qrs}$ to prevent the effect of respiration-related modulations of the R peak on morphSQ calculation.

The quantification of the signal quality as defined in step 11 can be perceived as a weighted peak NSR on percentage scale. Defining morphSQ as a NSR rather than a SNR increases the ease of use and interpretability. Lower values indicate a smaller amount of noise with a minimum of 0%, while at the same time the usage of the percentage scale is easier to interpret (even for inexperienced users) than the usual decibel/db scale for SNR. For a better understanding of morphSQ, Fig 3 shows different ECG samples and the corresponding morphSQ (additional cases are illustrated in S2 Fig). Using clean ECG recordings and realistic noise measurements from PhysioNet [45, 48, 49] we investigated the performance of beat detection and morphSQ in relation to signal noise. S3 Fig shows an example ECG from the Noise Stress Test database, where realistic noise was added to a clean record. morphSQ has been computed at each beat and the performance of beat detection in clean and noisy ECG sections was evaluated. The relation of average morphSQ with increasing SNR ratios was investigated using the methodology as described in [48]. S4 Fig summarizes the results and shows the direct relation to NSR. The similarity-based signal quality expressed by morphSQ is sensitive to beat misplacements and therefore requires accurate, complete, and verified beat annotations.

**B—Heartbeat detectability.** First, bandpass-filtering (3 Hz to 20 Hz) was applied on all signals to remove high frequency noise and baseline wandering but keeping the QRS complex. We applied several algorithms for beat detection (methods from WFDB toolbox [45, 50],

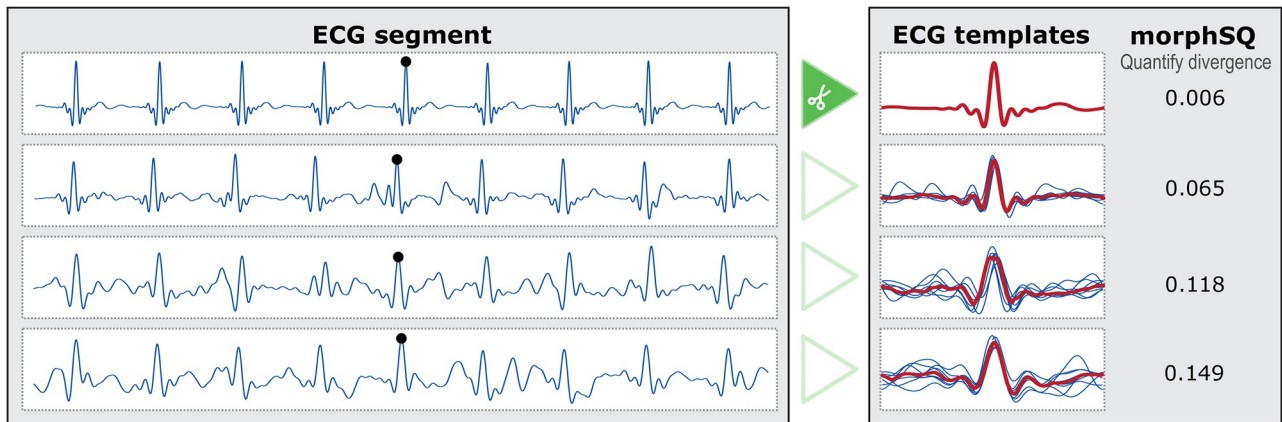

**Fig 3. Examples of ECG segments with different signal quality measured by quantifying the similarity of ECG templates.** Very clean segments have a morphSQ near to zero, noisy segments have values larger than 0.1.

HRVTool [51], RDECO [52], py-ecg-detectors [53], ECG2RR [54], and eplimited [44]) to search for the most accurate method and for the best channel in each device to detect heart beat locations. The open source algorithm eplimited [44], which is known to be less sensitive to noise [19], performed best in comparison to the screened methods on our experimental data and was further used to define the detected R peaks (beat$_{detected}$). If the signal quality is insufficient, the R peak detection either introduces false beat locations (FP—false positives) or some beat locations are missing (FN—false negatives).

 **C—Modification of the Smith-Waterman algorithm.** Polar RR interval sequences were generated by the manufacturer's in-house algorithm, with no ability to export the underlying ECG. In this case, the pairwise comparisons of beat$_{ref}$ with the exported data (query) is the direct way to assess the signal and export quality. For a fair comparison to the other ECG devices, eplimited was applied to the unsynchronized raw data (without correcting the sampling frequencies) to compute RR intervals. Since the interval data were not synchronized, a matching process had to be performed in order classify wrong or inaccurate RR intervals. In bioinformatics, the Smith-Waterman algorithm (SW) [55] is a well-known algorithm to perform local sequence alignment of nucleic acid sequences or protein sequences. We therefore decided to use a variation of SW to align sequences of numeric data types (RR intervals). A scoring matrix will be build that is based on gaps and individual ratings of matched pairs. The latter depends on the difference of matched RR intervals. Using dynamic programming and a traceback procedure, an optimal alignment can be derived. Thus, an open source implementation for global alignments (https://github.com/hiraethus/Needleman-Wunsch) was modified by redefinition of a match/mismatch and its respective reward or penalty values. A modifiable reward function was defined according to our needs to perform an alignment of RR interval sequences. Eqs 2 and 3 shows the definition of the scoring matrix $H$ based on two interval sequences $a$ and $b$ having $m$ and $n$ values respectively. Function $s$ of Eq 4 defines the reward of matched RR interval $a_i$ from the reference sequence to the RR intervals $b_j$ from the query sequence. Here we use a quadratic function based on the difference of both intervals in milliseconds. This defines the scoring for matched RR intervals allowing for minor deviations (reward = 1 if difference is 0 ms, reward = 0.9 if absolute difference is 10 ms, reward = 0.6 if absolute difference is 20 ms). The minimum value is capped by 0 and the maximum value is 1. Defining gap penalties as denoted in following Equations and in Fig 4, it was possible to find an optimal alignment of non-synchronized interval data. Interestingly, we have rewarded a gap in the reference sequence (an insertion) by 0.9 instead of penalizing it, since in our case this is always accompanied by a mismatch, which is already scored with 0. In this way, it was

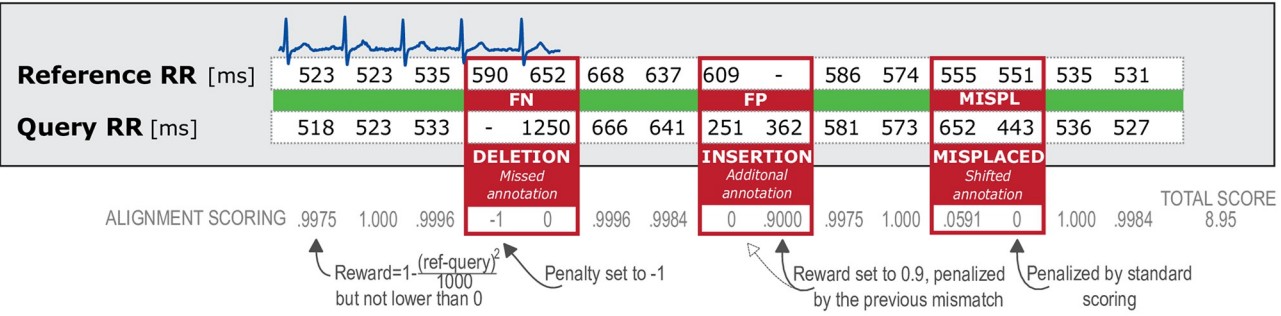

**Fig 4. Illustration of a Smith-Waterman-like algorithm for RR interval sequences to identify incorrect RR intervals in a query sequence.**

beneficial for the matching process and a better alignment could be obtained.

$$H(i, 0) = 0, \qquad i = 0, \cdots, m$$
$$H(0, j) = 0, \qquad j = 0, \cdots, n \tag{2}$$

$$H(i, j) = \max \begin{cases} 0 & \text{No similarity} \\ H(i-1, j-1) + s(a_i, b_j) & \text{Match/Mismatch} \\ H(i-1, j) - 1.0 & \text{Deletion} \\ H(i, j-1) + 0.9 & \text{Insertion} \end{cases}, \; 1 \le i \le m, 1 \le j \le n \tag{3}$$

$$s(a_i, b_j) = \max \begin{cases} 0 & \text{capped score} \\ 1 - \dfrac{(a_i - b_j)^2}{1000} & \text{RR interval deviation} \end{cases}, \; a_i \text{ and } b_j \text{ in ms} \tag{4}$$

An optimal alignment to maximize the overall score was found by the common traceback procedure. Using the resulting alignments of the ground truth RR intervals with the query sequences, we were to count and classify errors: gaps can be classified as either insertions (RR intervals broken up into smaller subintervals, usually treated as FP) or deletions (due to a missing beat location, usually treated as FN). Moreover, inaccurate RR intervals (due to misplaced beat locations) can be classified. Fig 4 illustrates possible findings by the matching process.

**Evaluation methodology.** The total amount of reference beats were computed based on the the reference annotation file (beat$_{ref}$) for the full experimental duration. For each experimental phase, total amount of beats, avgerage peak HR and median breathing rate (taken from Hexoskin) over all participant were computed.

The morphological signal quality index (morphSQI) was created by computing the mean and standard deviation (SD) of all morphSQ values as well as the rate of morphSQ < 10% for each ECG, participant, and experimental phase. For the summary of the complete experiment, we computed the mean over all participants along with the average SD. The choice of morphSQ values <10% to binary classify the overall signal quality is based on the descending performance of standard heart beat detection algorithms around this cut-off-value (see S3 Fig).

For heartbeat detectability the reference annotation was used in combination with a permitted deviation of 50 ms, to rate each beat that has been detected by eplimited either as positive (beat$_{detected}$ can be matched to the beat$_{ref}$) or negative (no reference beat location in the range of ±50 ms). The performance was expressed as positive predictive value (PPV) and false negative rate (FNR) [56]. Here, PPV is defined as the number of true positives (TP) divided by the total number of beats detected by eplimited. FNR is calculated as the number of FN divided by the sum of TP and FN. A time-window of 50 ms was regarded sufficient in heartbeat detection to allow post-processing algorithms for refinement which is stricter than the current gold standard of 150 ms [57]. Further, F1 scores were computed from PPV and sensitivity (true positive rate) in each of the probands' ECGs [58, 59]. The number of records with an F1 score below 0.90, between 0.90 and 0.99, and excellent scores greater than 0.99 were determined. In case of an F1 score lower than 0.90 the beat detection was regarded as insufficient which was caused by severe noise and artefacts. Insufficient beat detection may lead to the exclusion of

participant data to answer the intended research question. The accuracy was computed for the entire experimental period and each phase separately.

The SW process was performed with all extracted intervals from all used devices and all experimental phases separately to count deviations in the RR interval data. Again a 50 ms tolerance window was used. The summary table is showing the accumulated counts across all 13 participants.

**Data handling.** Data processing and statistical analysis was done in Matlab (MATLAB versions 9.6.0 to 9.9.0 (R2019a–R2020b), Natick, Massachusetts: The MathWorks Inc.), in R (v3.6.3 to v4.1.3 [60]), and in Python (v3.9.7).

## Results

The analysis of morphSQI was conducted with the complete dataset and in each of the four distinct phases (Table 3). Polar had to be excluded from the analysis, since no raw ECG was available. Overall, the differences between the four remaining devices are modest with morphSQI ranging between 0.023 (Faros) and 0.048 (Hexoskin). During the phases P2 and P4, participants were actively moving with elevated average peak HR of 103.3 bpm and 153.7 bpm respectively, and average median breathing rates of 24.1 and 27.1 cycles per minute (see Table 4). This resulted in a rise of morphSQI in all devices—with the smallest incline for NeXus and Faros while participants walked (P2). During the uphill phase (P4), morphSQI values for NeXus increased drastically to 0.072 with a high standard deviation among the different participants. Only the ECG recordings of Hexoskin achieved higher values (0.082).

Looking at the morphological signal quality, it is possible to gain insight into the noisiness on a single beat level. Table 3 displays the percentage of beats with a NSR below 10%. Faros had the lowest noise proportion over all phases, followed by SOMNOtouch. During P4 only 77.6% of all beats recorded using Hexoskin had a morphSQ below 10%. Besides Nexus, an increase in signal quality can be seen from P2 to P3 for all devices.

To rate the QRS detection performance, beat annotations for each device derived from eplimited (applied to bandpass-filtered signals) are compared to the reference annotation (beats$_{ref}$) with a permitted deviation of 50 ms. During the entire measurement period and over all participants, 42, 364 beats are observed. Within the relevant experimental phases, 27, 186 beats are recorded with small differences between P1 to P3 and more beats in P4 caused by increased HR through the moderate activity. The most FP and FN beats are detected using NeXus' ECG (678|758). These FPs and FNs can mostly be found in the uphill phase P4 (506| 491). NeXus is followed by Polar and Hexoskin regarding detection accuracy. Best results

**Table 3. Morphological Signal Quality Index (morphSQI) expressed as mean over all participants (average SD) and proportion of sufficient morphological signal quality (morphSQ < 10%).**

| | Overall | P1—Rest | P2—Walking | P3—2-back | P4—Uphill |
|---|---|---|---|---|---|
| **Mean (SD) morphological signal quality index (morphSQI)** | | | | | |
| eMotion Faros 360˚ | .023 (.011) | .018 (.006) | .026 (.006) | .016 (.006) | .016 (.006) |
| Hexoskin Hx1 | .048 (.036) | .024 (.008) | .070 (.016) | .022 (.008) | .082 (.025) |
| NeXus-10 MKII | .035 (.035) | .019 (.006) | .025 (.006) | .023 (.015) | .072 (.021) |
| SOMNOtouch NIBP | .028 (.020) | .016 (.006) | .042 (.009) | .016 (.006) | .038 (.009) |
| **Proportion of sufficient signal quality (morphSQ < 10%)** | | | | | |
| eMotion Faros 360˚ | 98.78% | 100.00% | 99.95% | 100.00% | 95.38% |
| Hexoskin Hx1 | 91.33% | 99.61% | 86.82% | 99.15% | 77.60% |
| NeXus-10 MKII | 96.48% | 100.00% | 98.88% | 97.51% | 89.10% |
| SOMNOtouch NIBP | 97.45% | 100.00% | 94.77% | 99.87% | 93.25% |

**Table 4. QRS detection performance expressed by false positive (FP) and false negative (FN) counts and positive predictive values (PPV) and false negative rates (FNR) from bandpass-filtered (3–20 Hz) ECGs (heartbeats annotated by eplimited).**

|  | Overall | P1—Rest | P2—Walking | P3—2-back | P4—Uphill |
|---|---|---|---|---|---|
| total reference beats | 42364 | 5623 | 6235 | 6106 | 9222 |
| avg. peak heart rate | – | 93.6 bpm | 103.3 bpm | 105.2 bpm | 153.7 bpm |
| avg. median breathing rate | – | 14.5 min⁻¹ | 24.1 min⁻¹ | 24.3 min⁻¹ * | 27.1 min⁻¹ |
| **FP\|FN using eplimited** |  |  |  |  |  |
| eMotion Faros 360˚ | 10\|123 | 0\|0 | 1\|0 | 0\|0 | 3\|1 |
| Hexoskin Hx1 | 393\|281 | 6\|1 | 171\|42 | 16\|0 | 128\|97 |
| NeXus-10 MKII | 678\|758 | 0\|0 | 1\|0 | 34\|21 | 506\|491 |
| SOMNOtouch NIBP | 117\|223 | 0\|0 | 40\|15 | 1\|0 | 11\|49 |
| Polar RS800 Multi | 532\|535 | 14\|17 | 40\|34 | 201\|203 | 1\|1 |
| **PPV\|FNR using eplimited** |  |  |  |  |  |
| eMotion Faros 360˚ | 1.000\|0.003 | 1.000\|0.000 | 1.000\|0.000 | 1.000\|0.000 | 1.000\|0.000 |
| Hexoskin Hx1 | 0.989\|0.008 | 0.999\|0.000 | 0.977\|0.007 | 0.997\|0.000 | 0.986\|0.011 |
| NeXus-10 MKII | 0.985\|0.017 | 1.000\|0.000 | 1.000\|0.000 | 0.994\|0.004 | 0.944\|0.055 |
| SOMNOtouch NIBP | 0.998\|0.005 | 1.000\|0.000 | 0.995\|0.002 | 1.000\|0.000 | 0.999\|0.005 |
| Polar RS800 Multi | 0.988\|0.012 | 0.997\|0.003 | 0.992\|0.007 | 0.970\|0.030 | 1.000\|0.000 |

* Respiration rate as measured by Hexoskin might be biased by speaking during the 2-back task.

could be observed using Faros. All devices achieve excellent results in resting state, missing no beats (Faros, NeXus, SOMNOtouch) or up to 0.6% (Polar). During movement phases, Faros keeps a low inaccuracy rate, SOMNOtouch shows a slight increase and NeXus had only significant deterioration during the uphill phase (P4) (10.8% misclassification rate = $\frac{FP+FN}{reference\ beats} = \frac{506+491}{9222}$). Polar has the highest percentage of missed beats during P3 (2-back) with 6.6% misclassification, but also the best result in P4 (0.02% misclassification) (see Table 4 for details).

PPV and FNR were used as a more intuitive approach for data comparison (Table 4). All devices achieved sufficient scores ranging from 0.985 (NeXus) to 1.000 (Faros). The comparatively low value for NeXus results from P4 (.944|.055). Faros is gaining best scores in all phases, SOMNOtouch is missing a few beats during movement phases. Polar scores better in phases with higher HR.

Further, F1 scores are used to describe accuracy of detected heartbeats on the participant level. Thus, the influence of single participants can be shown (Fig 5). An F1 score below 90% is considered as an insufficient ECG quality that results in increased error probability during automated beat annotation processes. This is the case for one participant measured with NeXus during P4 and in one participant measured with Hexoskin during walking (P2) but but not while running (P4). Overall, Polar had the most scores low or insufficient quality (below 99%: 7 participants), mainly attributable to the 2-back task, but achieves very high scores during the running phase. Using SOMNOtouch and Hexoskin one respectively three participants had low F1 scores.

For a better understanding and fair comparison to the Polar (RR interval) recordings, a modified SW algorithm was used to identify FPs, FNs, as well as misplaced beats from interval data that has also been extracted from unsynchronized raw ECGs computed from beat locations using eplimited. While for the results presented in Table 4 misplaced beats are usually counted both as one FP and one FN if the beat location differs by more than 50 ms, the

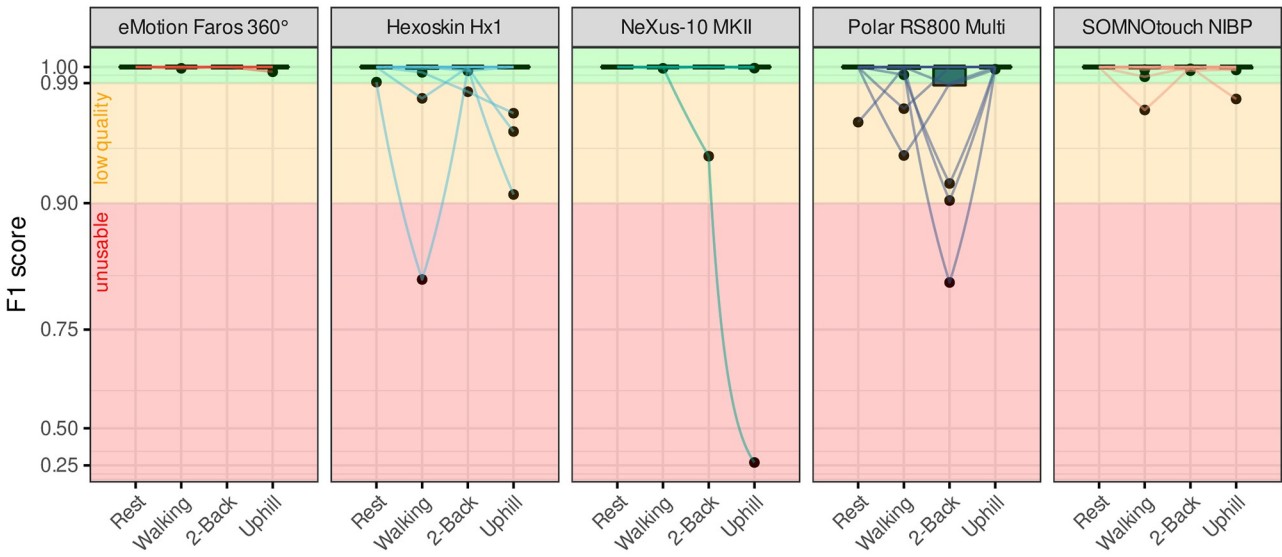

**Fig 5. QRS detection performance expressed by F1 scores of the used devices per experimental phase with individual curve progression.**

following analysis enables the distinction between misplaced beats, and FPs or FNs (Table 5). Insertions (respectively FP) and deletions (respectively FN) are less frequent than the number of misplaced beats, caused by noise around the QRS complexes. In resting conditions satisfying annotations were achieved in all recordings. The high numbers of misplaced beat especially in Hexoskin and NeXus were mainly caused by noise in data of a few participants as depicted in Fig 5 (please see S5 Fig including participant labels for a deeper insight).

## Discussion

Five ECG measurement devices made for specific use cases were tested. Holter ECG systems, chest straps, and biometric shirts all reached acceptable results with use case specific accuracy. This is supported by the low number of insufficient quality periods (see Fig 5). The medical device Faros achieved almost perfect results even during movement phases. Hexoskin's main problem is the dependency of the shirt size to fit the participant perfectly. Sensor placement may change during body movement and may result in reduced accuracy. NeXus overall accuracy is sufficient with deficits in the uphill condition. There is one participant for whom NeXus

**Table 5. QRS detection performance through RR interval alignment using the modified Smith-Waterman algorithm for numericals.** Errors are expressed by false positive counts (FP), false negative counts (FN), and the number of misplaced annotations from unsynchronized data (50 ms tolerance).

|  | Overall | P1—Rest | P2—Walking | P3—2-back | P4—Uphill |
|---|---|---|---|---|---|
| **FP\|FN\|Misplaced using Smith-Waterman approach** [*] | | | | | |
| eMotion Faros 360˚ | 5\|0\|19 | 0\|0\|0 | 0\|0\|0 | 0\|0\|0 | 0\|0\|6 |
| Hexoskin Hx1 | 85\|10\|765 | 3\|0\|5 | 114\|0\|204 | 11\|0\|14 | 29\|1\|270 |
| NeXus-10 MKII | 40\|5\|765 | 0\|0\|0 | 3\|0\|4 | 15\|0\|40 | 0\|1\|309 |
| SOMNOtouch NIBP | 63\|0\|202 | 0\|0\|0 | 8\|5\|134 | 0\|0\|2 | 0\|0\|65 |
| Polar RS800 Multi | 28\|29\|333 | 0\|3\|18 | 6\|0\|57 | 3\|3\|66 | 0\|0\|11 |

[*] Comparison based on exported Polar RR intervals and detected heart beats (eplimited) in raw unsynchronized ECGs.

produces unusable data (c.f. Fig 5). If this participant is excluded, NeXus reaches signal quality (morphSQI(P3) = 0.015, morphSQI(P4) = 0.028) comparable to Faros and SOMNOtouch. Polar, as the most popular mass-producing sports equipment manufacturer of the devices used, achieved the best results in phases with higher movement and thus higher HR. Nevertheless statistical differences has not been conducted due to the low number of participants.

A limitation of this study is the missing usage of a 12-lead ECG system as a gold standard for comparison. Such a system was not applicable due to possible interferences of the chest leads with the textile electrodes of Hexoskin. Clinically validated measuring devices were used in compensation. Additionally, the focus of this comparison was not on the application for the detection of clinical abnormalities, but on the application of the devices in ergonomics and sports science, as well as in the leisure context with the focus on the valid and automated detection of R peaks in order to reduce the effort involved in manual follow-up processing steps.

Measuring simultaneously with all devices enables the true comparison of the (morphological) signal quality, but at the same time restricts the perfect electrode placement for each device. Mutual interference was kept minimal by using Ag/AgCl electrodes. Several smaller discrepancies from manufacturer specifications for sampling frequency could be observed (e.g. NeXus using 8000 Hz instead of 8192 Hz) as well as unsteady fluctuations in sampling frequency especially for Faros and Polar. Using a single device, such variation would not be noticed nor corrected, as has been done in our approach [34]. This phenomenon, though, needs further investigation.

With some HRV indicators being really sensitive to missing or falsely detected beats it is important to check and correct automatically generated beat annotations. Especially for research purposes a device with accessible raw data should be used. While using devices storing just RR intervals, more robust indicators should be calculated (such as rrHRV [61]). To meet the current gold standard for HRV analysis, even with the best device and QRS detection algorithm, a visual inspection of annotated R peaks should be carried out by a professional to determine possible errors [36]. The accessibility to raw data is not always trivial. Most devices offer EDF format or other commonly used file types and users are able to access them via USB. For Hexoskin special software or skills are needed to extract the raw ECG from WAV format. For non-scientific usage, this can be challenging.

Wearables and similar self-quantification technologies are a fast moving and rapidly growing market [6]. With increasing signal quality, lower prices, and improvements in sensor technology, their future potentials are promising as long as they get validated. Healthcare can take advantage of it, especially through telemedical applications in rural areas. Promising approaches are made with ePatch to record data similar to 12-lead ECGs in more field like settings [62] and the Apple Heart Study shows first scientific possibilities to use consumer products for large scale clinical scenarios [63]. While most wearable HR detectors currently rely on PPG, this might change in the future with promising use cases from Apple Watch 4 and Withings Move. Future medical devices and consumer products will influence and enrich each other, but a technically and scientifically valid data acquisition should always be the basis of such development.

## Conclusion

Wearable ECG devices can be used as self-monitoring tools in leisure and mass sport, daily life, with special technical requirements for the use in scientific research as well as for the collection of big data sets in ambulatory settings. For work and sports related research, an affordable, valid device is a key aspect to allow for field studies. Although, medical studies in laboratory settings facilitate higher precision.

None of the used devices performed insufficiently overall. Most inter-device data-quality differences are only in nuances. A frequent usage in field settings requires the consideration of additional aspects: preparation time, mobility, and scientific usability. Preparation time is always dependent on the users' experience, but some limiting factors can remain the same. With just a chest belt to put on, Polar has the fastest mounting option, followed by the two devices with three electrodes (Faros and NeXus) and SOMNOtouch with four electrodes and the optional pulse oximeter. Hexoskin might take more time for the application of conductance gel and adjusting chest and abdomen straps afterwards. Nevertheless, all devices can be equipped in under five minutes. For mobility, handing of electrodes and cable placement is of importance, weight and other restrictive factors such as missing comfort or insufficient fit of textiles and strap bands have to be considered. Scientific usability further includes aspects of data handling, data access and interpretability. Sports devices offer a simple user experience and quick and easy interpretability of the data with no need for further raw data access which can be a limitation for research use cases. The remaining devices require experience in sensor placement and ECG data analysis to further interpret the data.

Accuracy still might be the most important aspect for research and ease of use in a scientific context. The data recorded during this study suggest that the clinically approved devices are the most accurate. Hexoskin's data quality relied on the fit of the shirts, with a loose fit causing the electrodes to move during physical activity resulting in an increased susceptibility to noise. With the missing possibility to extract the raw ECG, and compared to the other devices, Polar is the least accurate justified by the higher failure rate.

The simultaneous usage of five ECG devices differs from standard ECG use cases, but it offered the chance for a comprehensive comparison of data quality by two new approaches: analyzing the NSR using morphSQ, and RR interval quality using SW. Both methods offer intuitive and fast approaches to estimate data quality of a new device compared to a gold standard. morphSQ uses an intuitively interpretable percentage scale and works perfectly on a local level. It thus shows high correlation to SNR, FNR, and PPV of the detection of R peaks, but requires manually revised annotations of R peaks for reliable values. A future application of the morphSQ algorithm might be the additional analysis of p- and t-wave autocorrelation. Additionally to morphSQ, SW does not require synchronization to summarize and classify beat detection capability. There is also no need to handle incorrect sampling frequencies and signals interruptions during long-term measurements. In future research contexts, this approach can be applied to any sequences of numeric data, as the reward function can be easily adapted to other problems.

High scores for both indicators imply low manual postprocessing effort. Based on good signal quality, automated beat detection should lead to sufficient results. Thus, high morphSQ scores and good SW results indicate an increased ease of use/user friendliness during data processing and higher confidence in automatically generated/derived parameters.

Integrating a wearable device with ECG features into either leisure time activities or scientific research requires a careful consideration of available devices features and the corresponding task's demands. Some areas of application require more than a single-channel ECG or just RR intervals per se. Therefore, some devices are unsuitable. Before using new equipment, it is recommended to check the data quality thoroughly.

## Supporting information

**S1 Fig. General information about the measurement devices with images.**
(EPS)

**S2 Fig. Example of signal quality metric morphSQ with performance of beat detection in clean and noisy ECG segments.**
(PDF)

**S3 Fig. Part a) performance of beat-detection-algorithms in a noise stress test, part b) Signal quality metric morphSQ in relation to signal-to-noise ratio.**
(PDF)

**S4 Fig. Correlation of signal quality metric morphSQ with beat detection performance in noise stress test recordings.**
(PDF)

**S5 Fig. QRS detection performance expressed by F1 scores of the used devices per experimental phase with individual curve progression and participant labels.**
(PDF)

## Author Contributions

**Conceptualization:** Dominic Bläsing, Julian Elias Reiser.

**Data curation:** Marcus Vollmer.

**Formal analysis:** Dominic Bläsing, Marcus Vollmer.

**Funding acquisition:** Dominic Bläsing.

**Investigation:** Dominic Bläsing, Anja Buder, Julian Elias Reiser, Maria Nisser, Marcus Vollmer.

**Methodology:** Dominic Bläsing, Marcus Vollmer.

**Project administration:** Dominic Bläsing.

**Resources:** Dominic Bläsing, Anja Buder, Julian Elias Reiser, Maria Nisser, Marcus Vollmer.

**Software:** Marcus Vollmer.

**Supervision:** Dominic Bläsing.

**Visualization:** Marcus Vollmer.

**Writing – original draft:** Dominic Bläsing, Anja Buder, Julian Elias Reiser, Maria Nisser, Steffen Derlien, Marcus Vollmer.

**Writing – review & editing:** Dominic Bläsing, Anja Buder, Julian Elias Reiser, Maria Nisser, Steffen Derlien, Marcus Vollmer.

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
