## [Decision Letter · Decision Letter 0]

17 Dec 2021

PONE-D-21-31227Comparison of ECG signal quality, QRS detection performance, and device usability based on simultaneous recordingsPLOS ONE

Dear Dr. Blaesing,

Thank you for submitting your manuscript to PLOS ONE. After careful consideration, we feel that it has merit but does not fully meet PLOS ONE’s publication criteria as it currently stands. Therefore, we invite you to submit a revised version of the manuscript that addresses the points raised during the review process.

We look forward to receiving your revised manuscript.

Kind regards,

Thomas Martin Deserno, Ph.D.

Academic Editor

PLOS ONE

“The authors acknowledge the financial support by the Federal Ministry of Education and Research of Germany in the project Montexas4.0 (FKZ 02L15A261), travel grants from the German Centre for Cardiovascular Research (DZHK, partner site Greifswald), and the support for the Article Processing Charge from the DFG (German Research September 28, 2021 Foundation, 393148499) and the Open Access Publication Fund of the University of Greifswald.”

We note that you have provided information within the Acknowledgements Section. Please note that funding information should not appear in the Acknowledgments section or other areas of your manuscript. We will only publish funding information present in the Funding Statement section of the online submission form.

“DB acknowledges the financial support by the Federal Ministry of Education and Research of Germany in the project Montexas4.0 (FKZ 02L15A261 https://www.bmbf.de/bmbf/en/home/home_node.html), MV acknowledges travel grants from the German Centre for Cardiovascular Research (DZHK, partner site Greifswald https://dzhk.de/en/), all authors acknowledge the support for the Article Processing Charge from the DFG (German Research Foundation, 393148499 https://www.dfg.de/en/index.jsp) and the Open Access Publication Fund of the University of Greifswald.

Reviewers' comments:

Reviewer's Responses to Questions

**Comments to the Author**

1. Is the manuscript technically sound, and do the data support the conclusions?

Reviewer #1: Yes

Reviewer #2: Yes

Reviewer #3: Yes

2. Has the statistical analysis been performed appropriately and rigorously? 

Reviewer #1: N/A

Reviewer #2: Yes

Reviewer #3: Yes

3. Have the authors made all data underlying the findings in their manuscript fully available?

Reviewer #1: Yes

Reviewer #2: Yes

Reviewer #3: Yes

4. Is the manuscript presented in an intelligible fashion and written in standard English?

Reviewer #1: Yes

Reviewer #2: Yes

Reviewer #3: Yes

5. Review Comments to the Author

Reviewer #1: The authors present results of a study comparing signal quality of ECG recordings acquired using five commercially-available devices of which three are medical grade. Data was acquired in parallel as the devices employ different sensors technologies (e.g. conventional skin electrodes, textile sensors, chest strap) while the subjects (N=13) were performing different cognitive and physical activities. The dataset was already published within PhysioNet and preprocessing is described in a CinC 2019 contribution.

The topic is of high interest due to the increasing availability of (non-)medical grade ECG hardware - not only for interested individuals but also researchers - as it could enable ECG acquisition in scenarios that were not possible before. Evidently, the signal quality of these devices in different scenarios is a crucial point. The evaluation is based on a novel signal quality index that is proposed in this work. In conclusion, this work addresses an important topic and fits to PLOS ONE due to its multidisciplinary nature. The language of the paper is fine and the figures provided are of high-quality.

However, I have concerns regarding

i) the fact that this work introduces a novel signal quality measures without evaluating it or comparing it to other algorithms;

ii) the cited references as important up-to-date literature is not considered;

iii) the document content and structure being sometimes chaotic and not following IMRAD.

Moreover, iv) at multiple points in the work the wording is not uniform, the title requires changes and v) the level of detail is too superficial at some points.

However, I am optimistic that all issues can be addressed in an extensive revision, leading to acceptance.

i) The authors propose a new heuristic algorithm for ECG signal quality estimation. I am completely fine with that but this requires to evaluate its performance w.r.t. state-of-the-art methods in this field and/or test/synthetic signals with a pre-defined signal quality. Unfortunately, both are missing which prevents to assess the quality of the proposed method in any way.

I suggest the authors to either i) apply a state-of-the-art method to their data and compare the results to the results achieved with the proposed method or ii) apply their method to synthetic signals with manually added noise (e.g. baseline wander, muscle twitches) at pre-defined signal-to-noise levels (as done in [23]). Ideally, both experiments are conducted. This would lead the reader to appreciate the many open parameters (e.g. k=8, w=[-2,2], l_qrs=100ms, morphSQ < 10%) because at the moment the reader cannot assess how robust the proposed algorithm and results are.

Moreover, the reader cannot get "a feeling" for the proposed morphSQ values. Fig. 3 shows four ideal ECG signals and their corresponding morphSQ values but what about the special cases? What influence does baseline wander, motion artifacts etc. have on the accuracy of morphSQ?

ii) I strongly miss a "Related work" chapter in this work giving an overview of the state-of-the-art in ECG signal quality estimation. In the Introduction only two references are given which are from 2012 and 2007, respectively. I am absolutely not an expert in that field but I know some recent works that definitively have to be considered (e.g. https://doi.org/10.1109/tbme.2020.2969719 , https://doi.org/10.1109/rbme.2018.2810957) just by reading the relevant journals in that field.

iii) The document structure is chaotic at various points, especially in the first chapter:

ll. 1-14: Introduction and first mention of HR and HRV

ll. 15-23: "The aim of this study is to compare usability and data quality ..."

ll. 24-41: Fundamentals of ECG (clinical vs. Holter systems, PQRST etc)

ll. 42-50: The role of HR and HRV w.r.t. different user groups

ll. 51-60: Information on the devices used

ll. 60-68: "In this study, the ECG device comparison is based on..."

ll. 69-73: Information why a high signal quality is important in research

ll. 74-80: Short summary of ECG signal quality indices

ll. 81-85: "In this paper, a new signal quality index was developed"

As I try to underline, it is very hard for the reader to follow. I highly suggest to split the introduction into i) ECG fundamentals, ii) related work, iii) the aims of this work (or research questions), iv) and which methods will be used to reach these aims (answer the research questions).

At some other instances of the document, the authors do not follow IMRAD. For example:

- ll. 137-143: This should be raised in the introduction or discussed at the end of the work as a limitation.

- ll. 256-284: This chapter is hard to read as it mixes description of the algorithm and evaluation. I suggest to add a new subchapter "Evaluation methodology" (or similar).

- Conclusion: All the information on usability should be part of the main body of the text (method & results). The information how long a device needs to be equipped or how the authors assess the experience needed for sensor placement is not a "Conclusion" but a result.

iv) At multiple instances the wording is not uniform. Some examples are (this list is not complete):

- l. 22: "RR intervals" vs. l. 55: "RR-intervals"

- l. 54: "PPG" (not introduced)

- l. 10: "Heart Rate (HR)" vs. l. 49: "heart rate" (abbreviation not used)

- l. 129: "EDF or EDF+" (written in italics; also not introduced) vs. Table 2: "EDF" (no italics)

- l. 210: "PQT analysis" (what is that?) vs. l. 42: "PQRST"

- l. 259: "FP - false positive" vs. l. 312: "false positive (FP)"

Furthermore, the title does not contain the key information that in this work i) multiple, wearable ECG devices are compared and that ii) a novel method for signal quality estimation is proposed.

v) The description of the "Smith-Waterman-like algorithm" is too superficial. I cannot understand it from the 6 lines of text and the figure provided.

Minor issues:

- Please provide information how many annotators at what level of expertise provided beat_ref. Moreover, you report that arrhythmia periods were removed. Please report how this decision was made and how much data was removed.

- Table 1: That is a strange Euro symbol? Additionally, I think it would be good to report on the bit depth of the ECG signals.

- Table 2: Please define all abbreviations. What is EDF/SDC/ASC...?

- ll. 154-177: How long are P3 and P4?

- l. 202: I suggest to not use the wording "Statistical analysis" as no statistical testing is performed but only mean and SD values are reported.

- Abstract: "The results allow conclusions to be drawn..." -> I suggest to actually name the conclusions.

Reviewer #2: Summary

The authors compare five different ECG devices to evaluate these devices regarding QRS detection and usability. The research question is relevant. It is also important that the authors make the recorded data publicly available. However, it exists already many papers which compare different ECG devices. Before the publication, the method should be described sufficiently detailed, as some of the reasons for the selected approach are lacking. In addition, the limitations of the method should be discussed more intensively in the discussion.

Major Issues

• Introduction

o 1-4: The first reference should support this statement “[…] ongoing trend for self-quantification of personal health and fitness lead to […]” and is from 2012.

o 7-8: “A rising number of those devices are […]” here you need to add more than one reference.

o 23-41: It is more important to address wearable devices instead of the Holter ECG systems.

o 51-60: The selection of the different ECG devices belongs to the chapter “Material and Methods” and in the chapter “Introduction”.

• Material and Methods

o The authors need to address the time synchronization of the different devices. They need to explain how they can ensure that the recorded signals are the signals for the same time interval.

o Subjects: Please add the inclusion and exclusion criteria for the subjects.

o Statistical analysis: Please add the explanation why you chose the algorithm eplimited and did not choose another state-of-the-art algorithm (e.g., template matching or deep learning approach).

o 278: It is important to use the same algorithms to make the results comparable.

• Conclusion

o 401 „Fig 6. Authors' ordinal ranking of used recording devices.“: This ranking depends on the author's opinion and is not scientific. The authors need to exclude this figure.

Minor Issues

• Figures and tables

o Table 1: You should replace the euro signs with numbers.

o Figure 5: It is important to differentiate between the different use cases and to explain the results.

Reviewer #3: The authors present a study in which they compare several ECG devices for ambulatory monitoring in terms of their signal quality. The paper is very well written and the topic is of interest to the community. There are some points I would like to see addressed before publication.

- The authors introduce a novel signal quality index. I think it would make a lot of sense to either evaluate this signal quality index on existing benchmark problems first or to evaluate their data using also other signal quality indices and see whether or not there are any differences, if their new metric correlates with other metrics, etc. (as a side note, I find it somewhat unintuitive that the signal quality index has to assume low values for the signal to be of good quality. In Figure 3, the authors even write “quantify similarity”, so why not something like “1 – SQI”? Then, the authors also don’t need to talk about “Noise to Signal” in their manuscript but rather SNR...)

- The SQI cannot be applied to the Polar-device for obvious reasons. However, this makes the analysis somewhat inconsistent. Thus, I believe the authors should make a much stronger point on why to include this device (other than because it was there) or think about leaving it out.

- Along the same lines, the authors make an argument as to why they are not using a 12-lead ECG as reference (which is plausible) while, at the same time, argue that their SQI might be useful in the context of “further PQT analysis”. This seems like an ambiguity to me.

- I find the presentation of results in the large tables somewhat confusing. The authors could think about, for example, highlighting the best values, etc. Also, is there a way to perform some sort of statistical test to see whether or not the results of the different devices and test scenarios are significantly different?

- The conclusion section is rather long and also introduces a new figure, I would consider rearranging the content between discussion and conclusion.

- The paper could benefit from another round of proof reading from a native speaker ”differences in usability and practicability between all devices ARE characterized”, “As it was the goal to induce a mental state” (?). Also, it seems to me that punctuation could be improved.

- “During a first check of the recorded data from all five devices, several problems occurred that needed to be solved.” I am not sure this is a good sentence to start the paragraph as this should be a result-oriented (and not a progress-oriented) report.

6. PLOS authors have the option to publish the peer review history of their article (what does this mean?). If published, this will include your full peer review and any attached files.

Reviewer #1: No

Reviewer #2: No

Reviewer #3: No

---

## [Author Response · Author response to Decision Letter 0]

16 Jun 2022

Dear Reviewers, 

thank you all for taking your time and providing us with your thoughtful comments. We are confident that our revised manuscript benefited from your input. Please find our detailed response in the added PDF. Supplemental figures A1-A4 are only for this response letter and should not be published with the manuscript. 

Yours sincerely

---

## [Decision Letter · Decision Letter 1]

16 Aug 2022

PONE-D-21-31227R1ECG performance in simultaneous recordings of five wearable devices using a new morphological Noise-to-Signal index and Smith-Waterman-based RR interval comparisonsPLOS ONE

Dear Dr. Blaesing,

Thank you for submitting your manuscript to PLOS ONE. After careful consideration, we feel that it has merit but does not fully meet PLOS ONE’s publication criteria as it currently stands. Therefore, we invite you to submit a revised version of the manuscript that addresses the points raised during the review process.

We look forward to receiving your revised manuscript.

Kind regards,

Thomas Martin Deserno, Ph.D.

Academic Editor

PLOS ONE

Journal Requirements:

Additional Editor Comments (if provided):

I suggest to consider the paper metioned by reviewer 2

Reviewers' comments:

Reviewer's Responses to Questions

**Comments to the Author**

1. If the authors have adequately addressed your comments raised in a previous round of review and you feel that this manuscript is now acceptable for publication, you may indicate that here to bypass the “Comments to the Author” section, enter your conflict of interest statement in the “Confidential to Editor” section, and submit your "Accept" recommendation.

Reviewer #1: All comments have been addressed

Reviewer #3: All comments have been addressed

2. Is the manuscript technically sound, and do the data support the conclusions?

Reviewer #1: (No Response)

Reviewer #3: Yes

3. Has the statistical analysis been performed appropriately and rigorously? 

Reviewer #1: (No Response)

Reviewer #3: N/A

4. Have the authors made all data underlying the findings in their manuscript fully available?

Reviewer #1: (No Response)

Reviewer #3: Yes

5. Is the manuscript presented in an intelligible fashion and written in standard English?

Reviewer #1: (No Response)

Reviewer #3: Yes

6. Review Comments to the Author

Reviewer #1: Dear authors, thank you very much for the thorough revision of your manuscript. I congratulate you to this effort and suggest to highlight this work on the PLOS ONE website.

Reviewer #3: The authors addressed all my comments (and I believe those of the other reviewers as well) sufficiently and have improved the manuscript significantly in the process. I now find the manuscript acceptable for publication.

Looking at the not super-recent literature, this paper [https://doi.org/10.3390/s18010038] introduces a “shape-based signal-to-noise ratio SNR_S” for cardiorespiratory signals that, to me, seems to have some obvious connections to the “morphological signal quality index” presented here. Thus, the authors might want to think about discussing it in the related work section.

7. PLOS authors have the option to publish the peer review history of their article (what does this mean?). If published, this will include your full peer review and any attached files.

Reviewer #1: **Yes: **Nicolai Spicher

Reviewer #3: No

---

## [Author Response · Author response to Decision Letter 1]

26 Aug 2022

Dear Reviewers, 

thank you for taking the time to review our revised version of the manuscript. We are pleased that our changes have met your expectations and that you now consider the article worthy of publication.

@Reviewer 3: Please find our answer to your suggestion in the "Response to Reviewers" File. We read the suggested reference and included it in our manuscript.

---

## [Editor Report · Decision Letter 2]

9 Sep 2022

ECG performance in simultaneous recordings of five wearable devices using a new morphological Noise-to-Signal index and Smith-Waterman-based RR interval comparisons

PONE-D-21-31227R2

Dear Dr. Blaesing,

We’re pleased to inform you that your manuscript has been judged scientifically suitable for publication and will be formally accepted for publication once it meets all outstanding technical requirements.

Kind regards,

Thomas Martin Deserno, Ph.D.

Academic Editor

PLOS ONE
---

## [Editor Report · Acceptance letter]

26 Sep 2022

PONE-D-21-31227R2 

ECG performance in simultaneous recordings of five wearable devices using a new morphological Noise-to-Signal index and Smith-Waterman-based RR interval comparisons 

Dear Dr. Blaesing:

I'm pleased to inform you that your manuscript has been deemed suitable for publication in PLOS ONE. Congratulations! Your manuscript is now with our production department. 

Kind regards, 

on behalf of

Dr. Thomas Martin Deserno 

Academic Editor

PLOS ONE